# Pathological Changes in the Lungs of Patients with a Lethal COVID-19 Clinical Course

**DOI:** 10.3390/diagnostics12112808

**Published:** 2022-11-15

**Authors:** Valters Viksne, Ilze Strumfa, Maris Sperga, Janis Ziemelis, Juris Abolins

**Affiliations:** 1Pathology Center, Riga East Clinical University Hospital “Gaiļezers”, Hipokrata Street 2, LV-1038 Riga, Latvia; 2Department of Pathology, Riga Stradins University, 16 Dzirciema Street, LV-1007 Riga, Latvia

**Keywords:** COVID-19, diffuse alveolar damage, autopsy, histology, thrombosis, lung infarction

## Abstract

The novel coronavirus SARS-CoV-2 was identified in 2019 and quickly became the cause of the fifth worst pandemic in human history. Our goal for this research paper was to examine the morphology of the lungs in 88 patients that died from COVID-19 in Latvia, thus increasing the data available about the histological characteristics of SARS-CoV-2-induced disease. Lung tissue samples from 88 autopsies were visualized in hematoxylin–eosin and assessed by light microscopy. The male-to-female ratio was 56:32, and the mean age was 62 years ± 15.5 years (22–94 years). Clinically important laboratory data were assessed, including leucocyte count, CRP (C-reactive protein) and D-dimer levels. Signs of diffuse alveolar damage were found in 83/88 (94.3%; 95% CI 87.0–97.9) of patients, 38/88 (43.2%; 95% CI 33.3–53.6) in the exudative phase, and 45/88 (51.1%; 95% CI 40.8–61.3) in the proliferative phase. Vascular damage was identified in 70/88 (79.5%; 95% CI 69.9–86.7) of patients, and 83/88 (94.3%; 95% CI 87.0–97.9) had signs of thrombosis. A sparse inflammatory infiltrate of lymphocytes and macrophages was a common finding aside from cases with an identified coinfection. Eighty patients had significant co-morbidities, including coronary heart disease (49), primary arterial hypertension (41), and diabetes mellitus (34). Since our group’s demographic profile and spectrum of co-morbidities were analogous to other reports, the histological findings of marked diffuse alveolar damage, widespread vascular lesions, and active thrombosis can be considered representative of severe COVID-19.

## 1. Introduction

Toward the end of 2019, the first cases of a new type of coronavirus were identified in Wuhan, China. The virus was named (severe acute respiratory syndrome coronavirus 2 (SARS-CoV-2) due to its close resemblance to previously discovered coronaviruses severe acute respiratory syndrome coronavirus (SARS-CoV) and Middle East respiratory syndrome coronavirus (MERS-CoV). The disease, subsequently named coronavirus disease 2019 (COVID-19), caused the fifth worst pandemic in human history in March 2020. COVID-19 has infected more than 474 million people thus far, resulting in 6.1 million deaths. However, the number of studies analyzing autopsy findings represents a fraction of overall cases. In October 2020, Maiese et al. [1] published a literature review that analyzed 28 scientific papers totaling 341 cases. In June 2022, Hammoud et al. [2] reviewed 50 articles totaling 430 cases. Morphological findings are vital to understanding COVID-19 pathogenesis. Our goal for this paper was to supplement the data available on common morphological patterns caused by COVID-19 by analyzing the pathological changes found in 88 patients with a lethal disease course in Latvia.

## 2. Virology of COVID-19

SARS-CoV-2 is an enveloped, oval virus roughly 60–100 nm in diameter. Its structure is similar to that of other coronaviruses, such as SARS-CoV and MERS-CoV, which were discovered in 2003 and 2012, respectively. The natural reservoir of both of these viruses is bats; therefore, it is likely that SARS-CoV-2 shares the same origin. However, the actual transmission mechanism to humans is still unknown [3,4].

Like other β-coronaviruses, SARS-CoV-2 has a nucleocapsid that contains phosphorylated nucleocapsid (N) proteins and positive-stranded RNA with a genome size of roughly 30 kbp. The envelope comprises matrix (M) and envelope (E) proteins. On the surface of the envelope, two types of spike proteins can be found: the hemagglutinin-esterase (HE) and spike (S) glycoprotein trimer. The latter is responsible for binding to the human angiotensin-converting enzyme 2 receptor (ACE2 receptor) found in numerous human tissues [3]. In the heart, it is mostly localized on the surface of endothelial cells in the capillaries of the myocardium, medium caliber coronary arteries, and arterioles and, to a lesser extent, in the smooth muscle cells and adventitia of large arteries. The ACE2 receptor in the kidneys is found on the surface of endothelial cells, smooth muscle cells, and the epithelium of the proximal tubule [5]. In the lungs, the receptor is commonly found in both the epithelium and endothelium. The widest distribution is found on the apical surface of the ciliated epithelium of the lung parenchyma [6].

The S proteins of coronaviruses exist in a partly stable pre-fusion conformation and comprise two subunits: S1 and S2. Both undergo conformation changes so the viral membrane can fuse with the target cell. The structural rearrangement begins when the S1 subunit binds to the ACE2 receptor and destabilizes the spike protein, which results in the shedding of the S1 subunit by proteolysis. Consequently, the S2 subunit transitions to a highly stable conformation [4]. After entering the target cell, the viral RNA is integrated into the host cell’s ribosomes, serving as a matrix for new viral protein synthesis. SARS-CoV and SARS-CoV-2 infect cells using this mechanism, but the affinity of SARS-CoV-2 to the ACE2 receptor is about 10–20 times higher than that of SARS-CoV. This finding may explain why human-to-human transmission of COVID-19 is so prominent compared to other coronaviruses [4,7].

## 3. Lung Pathology: Morphological Differential Diagnosis of COVID-19

COVID-19 may be responsible for multiple pathological changes in the patient’s lungs. The following is a characterization of the most common morphological findings.

Respiratory failure caused by acute, bilateral lung damage is clinically defined as acute respiratory distress syndrome (ARDS). According to the Berlin definition published in June of 2012 [8], ARDS can be diagnosed when cardiogenic pulmonary edema and alternative causes of acute hypoxemic respiratory failure are excluded and radiologically if patients have bilateral lung infiltrates with no alternative cause and respiratory symptoms developed within a week. The main clinical diagnostic criterion of ARDS is an impairment of oxygenation, defined by the ratio of partial arterial oxygen pressure to a fraction of inspired oxygen (PaO_2_/FiO_2_) below 200 mmHg [8]. Histologically, most of these and severe COVID-19 pneumonia patients will develop diffuse alveolar damage (DAD), the most severe morphological manifestation of ARDS [9]. Other ARDS manifestations include acute eosinophilic pneumonia (AEP) and acute fibrinous and organizing pneumonia (AFOP) [10].

DAD is divided into three distinct histological phases: the acute or exudative phase, the subacute or organizing/proliferating phase, and the chronic or fibrotic phase. The exudative phase develops within a week of the initial lung damage and is characterized by intra-alveolar and interstitial edema and hyaline membranes rich in eosinophilic protein and detritus. These membranes line or replace the alveolar septa. Inflammation in this phase is sparse and mostly consists of mononuclear cells (macrophages and lymphocytes), with neutrophils sometimes present in low numbers. Vascular damage, characterized by micro thromboembolism and hemorrhage, is commonly found in this phase. The subacute or organizing phase typically develops a week after the acute phase and is characterized by fibrin, collagen deposits, and fibroblast proliferation. Hyaline membranes are organized into the new connective tissue; therefore, they are rarely seen. In addition, type 2 pneumocyte hyperplasia and potential squamous metaplasia can develop. Many cases of DAD will resolve following this phase with no lasting damage; however, some patients will progress to the chronic phase, which is characterized by widespread fibrosis similar to usual interstitial pneumonia found in many interstitial lung diseases [9,10,11].

AFOP and AEP are the main histological differential diagnoses of DAD. AFOP is characterized by acute lung damage, which deposits a large amount of fibrin in the alveoli, but no hyaline membranes form. Lymphocytes are found in patients with AFOP, but neutrophils and eosinophils should not be seen. Therefore, to diagnose AFOP, a large biopsy sample is required to exclude common DAD findings (hyaline membranes), AEP (eosinophils), and bacterial pneumonia (neutrophils). In AEP, fibrin deposits are also found, but the inflammatory infiltrate consists of macrophages and eosinophils in both the interstitial space and alveoli [9,10].

It is important to distinguish the morphology of primary COVID-19 pneumonia, which manifests with DAD, and secondary bacterial pneumonia. The latter are classified into two types, lobar and bronchopneumonia, but in most clinical cases, findings of both types can overlap. The progression of lobar pneumonia is further divided into four phases: congestion, red hepatization, gray hepatization, and resolution. The first phase is characterized by vascular congestion, extensive intra-alveolar edema, the presence of bacteria, and few neutrophils. Red hepatization is characterized by massive exudation in the alveoli comprising fluid, neutrophils, fibrin, and erythrocytes. Gray hepatization manifests with the disintegration of erythrocytes and a fibrinous, pus-filled exudate in the alveoli. Finally, in the resolution phase, the intra-alveolar exudate is enzymatically digested by macrophages or organized by fibroblasts growing into it. In bronchopneumonia, the infiltrates are multifocal and patchy, consisting of suppurative inflammation, fibrin exudation, and neutrophil infiltration in the bronchi, bronchioles, and adjacent alveoli [12].

## 4. International Research on Lung Pathology in COVID-19 Patients

The first publications on the morphological changes caused by COVID-19 in the lungs were published in February 2020. Xu et al. [13] analyzed autopsy material of the lungs of a 50-year-old COVID-19 patient who died nine days after the disease onset. Histologically, the researchers found pathological changes that resembled those caused by SARS and MERS in other patients. Early signs of DAD were pneumocyte desquamation, hyaline membrane formation, and an interstitial mononuclear inflammatory cell infiltrate in which lymphocytes dominated. The researchers also found syncytial multinucleated giant cells and atypical pneumocytes with an enlarged nucleus, granular cytoplasm, and prominent nucleoli. The latter changes were attributed to the cytopathic effect caused by the virus; however, unequivocal signs of intracellular viral particles were not found [13].

Tian et al. [14] analyzed the lung material of two patients who underwent a lobectomy due to adenocarcinoma. Prior to the operation, neither patient had signs of a respiratory infection. However, it developed after the operation, and both patients had COVID-19. The researchers found vascular congestion and protein-rich fluid in the alveoli. The inflammatory infiltrate was sparse, including macrophages and multinucleated giant cells. Signs of fibroblast and type 2 pneumocyte proliferation were also present, which the researchers characterized as early signs of DAD. However, hyaline membranes and vascular damage were not found [14].

In April, Varga et al. [15] published an article that provided well-justified evidence of SARS-CoV-2 tropism in endothelial cells. The researchers examined the inner organs (lungs, kidneys, heart, and small bowel) of three patients with electron microscopy. They found apoptotic bodies and intracellular viral particles in these cells in all four organs. The likely reason for this finding is the ACE2 receptor itself, which is expressed diffusely by multiple cell types. Nevertheless, the most prominent COVID-induced morphological changes were still located in the respiratory system [15].

In May, Barton et al. [16] published two cases illustrating a fundamental issue examining lung material affected by COVID-19 post-mortem. The first patient, a 77-year-old man with multiple co-morbidities, died 6 days after disease onset. In the autopsy material, classic signs of DAD were found, including hyaline membranes, focal, sparse mononuclear inflammatory cell infiltrate, and vascular damage characterized by micro-thromboembolism and hemorrhage. However, the researchers found signs of acute bronchopneumonia in the second patient, a 42-year-old man with myotonic muscular dystrophy. Even though the patient was positive for COVID-19, prominent neutrophil infiltration and aspiration signs (food particles and bacteria) were found. Therefore, it was likely that the patient died from aspiration pneumonia and not COVID-19 [16]. The wide spectrum of differential diagnoses for acute lung damage can complicate detecting the true cause of death in COVID-19 patients because several have an increased risk of developing hospital-acquired lung infections. Critically ill patients often spend weeks in intensive care units while intubated.

The largest autopsy study of COVID-19 until that point was conducted by Menter et al. in May [17], which included 21 patients at a mean age of 76. Among them were 17 males and 4 females. All patients had multiple co-morbidities, including primary arterial hypertension, cardiovascular diseases, diabetes mellitus, and other factors that could increase the risk of a severe disease course, e.g., smoking and obesity. This study also examined the macroscopic findings of the lungs. Most patients had focal and uneven lung consolidates, but some had diffuse, suppurative infiltrates that resembled bronchopneumonia. Microscopically, all (100%) patients had vascular congestion and 76% had the exudative phase of DAD, but 38% had the proliferative phase of DAD. The most common DAD findings were hyaline membranes, micro-thromboembolism, intra-alveolar fibrinous exudate, and focal lymphoid inflammatory infiltrate. The researchers also found a bacterial coinfection in 10 patients. Six of these had diffuse, and four had focal bronchopneumonia. In terms of vascular damage, only three patients had hemorrhages in the lung parenchyma, and immunohistochemical visualization of fibrin in 11 patients revealed thromboses in only five [17]. 

Following a trend of larger patient groups, Carsana et al. [18] published a paper on 38 patients in June. DAD in the exudative phase was found in all patients, whereas a sparse mononuclear inflammatory infiltrate was identified in 82% of cases, which continued to be consistent with previous research. Vascular damage was also more pronounced, with 87% of patients having small vessel thromboses. However, fibrosis was a more common finding in this study than in previous data. Mural fibrosis was found in 63% of cases, and 39% of patients had a microcystic honeycombing pattern of fibrosis even though no patient had a clinical history of pre-existing interstitial lung disease [18].

One of the most comprehensive studies of histopathological changes in COVID-19 patients was published in April 2021 by Bryce et al. [19], who reported on 100 autopsy cases. The researchers examined multiple organ systems both macroscopically and microscopically. Gross findings of the lungs displayed either firm, tan consolidation with abundant edema or diffusely firm and solid tissues. The morphological changes were consistent with findings by other researchers displaying DAD in 82 patients, of which 54 were in the acute phase, and 28 were in the organizing phase. The morphological patterns and criteria for the DAD phase were also consistent. In the study, multiple panels of immunostains were performed first to identify the cell populations of a generally mild inflammatory infiltrate. CD4+ T cells and CD163+ macrophages were predominant. CD61 stains were used to identify the presence of thrombi in small vessels and were found in 21 of 23 patients. Finally, the study reported on a fairly large amount of superimposed acute pneumonia, which was present in 45 patients [19].

## 5. Materials and Methods

We analyzed 88 autopsies performed on COVID-19-positive patients in Riga East Clinical University Hospital “Gaiļezers”. The autopsies were conducted according to the modified en masse (*modo Letulle*) method [20]. Initially, the dead body was inspected macroscopically, body cavities were opened, and the inner organs were evacuated. The complex of organs was evaluated on an autopsy table; each organ was measured and weighed, and a representative tissue sample(s) was obtained. The samples were then fixated in formalin, dehydrated in a tissue processor, and embedded in paraffin [21]. We performed the visualization of samples in hematoxylin–eosin with light microscopy. 

Anonymized demographic and clinical parameters were gathered. These included the patient’s gender, age (years), length of hospital stay (days), length of disease (days from the onset of symptoms to death), and co-morbidities (count; frequency, %; 95% CI). Gender and co-morbidity data were evaluated as qualitative categorical parameters, and age, length of hospital stay, and length of disease as quantitative data. Clinically important laboratory findings that could potentially correlate with morphological data were also compiled. These included leukocyte count (reference range: 4–8.8 × 10^3^/μL), C-reactive protein (reference range <5 mg/L), and D-dimer levels (reference range <0.5 μg/mL). The most likely values that could correlate with morphological data were assumed to be the last available ones prior to the patient’s death; therefore, these were used for evaluation. 

Lung tissue samples were evaluated under light microscopy. All parameters were chosen according to data available in the literature about the most common findings in COVID-19 and other viral types of pneumonias [9,10,11]. The diagnosis of DAD was established by examining the following qualitative categorical binary parameters: diffuse alveolar damage, exudative phase or proliferative phase; vascular damage (micro thromboembolism and hemorrhage), large and medium caliber blood vessel thromboembolism, interstitial and intra-alveolar fibrosis, and hemorrhagic infarction. The infiltration density of each inflammatory cell (neutrophil, macrophage, and lymphocyte) was evaluated as a semi-quantitative parameter (cells absent; scant cells; intermediate number of cells; high density of the respective cell). Finally, we also evaluated the presence of pulmonary embolism (PE) and multinucleated giant cells as qualitative categorical binary parameters. Statistical analysis was performed, including descriptive and analytical assessment. For continuous data, the mean value and standard deviation (SD) were detected. The data were shown as the mean ± SD (range). The occurrence of categorical parameters was characterized by frequency (%) and the corresponding 95% confidence interval (CI). For analytical assessment, the non-parametrical Mann–Whitney U test was applied. Any differences were considered statistically significant if *p* < 0.05. The IBM SPSS Statistics software (manufacturer IBM, Armonk, NY, USA) was used for data evaluation. 

## 6. Results

Of the 88 patients in our study, 56 were male, and 32 were female, with a mean age of 62.2 ± 15.5 years (22–94 years). The mean length of hospital stay was 10.4 ± 9.8 days (0–39 days), and the mean length of disease was 15.5 ± 10.1 days (0–47 days). The most common co-morbidities were coronary heart disease (49/88 patients; 55.7%; 95% CI 45.3–65.6), primary arterial hypertension (41/88; 46.6%; 95% CI 36.5–56.9), adiposity (41/88; 46.6%; 95% CI 36.5–56.9), diabetes mellitus (34/88; 38.6%; 95% CI 29.1–49.1), chronic kidney disease (24/88; 27.3%; 95% CI 19.0–37.4), and chronic obstructive pulmonary disease (3/88; 3.4%; 95% CI 0.7–9.9). Multiple patients had unique co-morbidities, including schizophrenia (1/88; 1.1%; 95% CI 0.0–6.8), bronchial asthma, HIV, and others. Only eight patients (8/88; 9.1%; 95% CI 4.5–17.2) had no identifiable co-morbidities.

Data on leukocyte counts were available for 84 patients with a mean of 14.6 ± 8.9 × 10^3^/μL (0.7–51.5). Among them, 61 had leukocytosis, 4 had leukopenia, and 19 had a normal leukocyte count. C-reactive protein levels were available for 85 patients and were elevated in 83 cases with a mean of 177.2 ± 126.3 mg/L (0.8–505.4). Multiple patients had either clinical or histological confirmation of bacterial or fungal co-infections. Clinically, one patient’s sputum culture revealed Pseudomonas aeruginosa 10^5^ colony-forming units (CFU)/mL and Stenotrophomonas maltophilia 10^5^ CFU/mL. Another patient had Acinetobacter baumanii 10^3^ CFU/mL and Candida albicans 10^5^ CFU/mL in the bronchoalveolar lavage fluid, and yet another patient had different Staphylococcus species identified in the sputum culture.

Of the 76 patients for whom D-dimer levels were available, 71 had an increase (0.7–71.3 μg/mL), 2 had a borderline increase (0.52 and 0.54 μg/mL) and only 3 had normal levels (0.42–0.48 μg/mL). The mean D-dimer level was 9.1 ± 12.5 μg/mL (0.4–71.3). A summary of the clinical parameters is presented in Table 1.

Histologically, 83/88 (94.3%; 95% CI 87.0–97.9) of patients had diffuse alveolar damage (DAD). Of these, 38/88 (43.2%; 95% CI 33.3–53.6) were in the exudative phase, as evidenced by collapsed alveolar structures, pneumocyte desquamation, hyaline membranes, and a sparse inflammatory infiltrate (Figure 1a). Patients totaling 45/88 (51.1%; 95% CI 40.8–61.3) of cases had DAD in the proliferative phase, as characterized by the presence of interstitial and intra-alveolar fibrosis, a sparse inflammatory infiltrate with more macrophages and a lack of hyaline membranes. Vascular damage was identified in 70/88 (79.5%; 95% CI 69.9–86.7) of cases, as evidenced by intra-alveolar hemorrhage and micro thromboembolism (Figure 1a and Figure 2b). Small, medium and large caliber vessel thromboses were identified in 83/88 (94.3%; 95% CI 87.0–97.9) of cases, with 12/88 (13.6%; 95% CI 7.8–22.5) of patients having developed a pulmonary embolism (PE) (Figure 2a,b). Hemorrhagic infarctions were identified in 27/88 (30.7%; 95% CI 22.0–41.0) of patients.

The inflammatory infiltrate was evaluated by the density of each inflammatory cell (neutrophil leukocyte, macrophage, and lymphocyte) and the presence of multinucleated giant cells. Almost all of the patients had inflammation in the interstitial and intra-alveolar spaces (Figure 1a,b) but rarely in the bronchial lumen. In 47/88 (53.4%; 95% CI 43.1–63.4) of cases, neutrophils were not identified. However, macrophages and lymphocytes were found in almost every case (Figure 1a,b). Macrophages were present in intermediate and high density in 42/88 (47.7%; 95% CI 37.6–58.0) and 11/88 (12.5%; 95% CI 7.0–21.2) of cases, respectively. Scant lymphocytes were present in 47/88 (53.4%; 95% CI 43.1–63.5), and an intermediate amount in 26/88 (29.5%; 95% CI 21.0–39.8) of cases. Multinucleated giant cells were found in 60/88 (68.2%; 95% CI 57.8–77.0) of cases. Abscesses and signs of bacterial pneumonia were identified in eight cases, with another two having histologically confirmed fungal co-infections (Figure 3a,b). All 10 patients had dense neutrophil leukocyte infiltrates. It is worth noting that co-infections were clinically confirmed in three patients, as mentioned in a previous paragraph. A summary of the morphological data on COVID-19 patients’ lung tissue samples is shown in Table 2.

For statistical analysis, certain parameters were compared for correlation using the Mann–Whitney U test. A statistically significant correlation was identified between the length of disease and exudative phase DAD (mean = 10.2 ± 7.6 days) and proliferative phase DAD (mean = 21.1 ± 9.5 days, *p* < 0.001). The same test was used to compare hemorrhagic infarctions with length of disease and D-dimer levels. Neither test found a statistically significant correlation (*p* = 0.128 and *p* = 0.273, respectively).

## 7. Discussion

This autopsy series continues to expand on the morphological findings of lung damage caused by COVID-19. In all but five cases, patients had DAD with a slight predominance toward the proliferative phase of DAD, which signifies a later stage in the disease. The morphology of inflammation was also consistent with DAD: mononuclear cell infiltrates (lymphocytes and macrophages) were predominantly found at low or intermediate density. Vascular damage and thrombosis were identified in 70 and 83 patients, respectively. Other research groups have reported similar findings in Europe and elsewhere [14,16,17,18].

Respiratory failure due to COVID-19 pneumonia was the direct cause of death in every patient. Since DAD is the dominant morphological pattern in these patients, it is the most likely culprit of respiratory failure. However, DAD is not specific to any diagnosis. In both severe acute respiratory syndrome (SARS) and Middle East respiratory syndrome (MERS), diffuse alveolar damage with hyaline membranes, edema, and interstitial lymphocytic pneumonia are frequently found [22]. Similarly, vascular damage is a common finding in SARS patients, as evidenced by Hwang et al. [23] on 20 patients. This study found hemorrhages in 80% of patients with a disease length of <14 days and hemorrhagic infarctions in 73% of patients with a disease length of >14 days. Morphological changes caused by influenza can also occasionally be similar to COVID-19. In 2011, Calore et al. [24] analyzed lung autopsy samples from six patients who died due to H1N1 infection. In five cases, DAD was found with its typical characteristics of hyaline membranes, pneumocyte desquamation, and multinucleated giant cells. Similarly, Capelozzi et al. [25] and Harms et al. [26] found DAD and widespread intra-alveolar hemorrhages in all cases. For these reasons, DAD is the likely cause of respiratory failure and death in any patient with a severe clinical course of viral pneumonia.

Multiple research groups have reported a high incidence of vascular damage caused by COVID-19 [18,27,28]. Varga et al. [15] described the likely reason for this by reporting on direct endothelial damage caused by SARS-CoV-2 using electron microscopy. The findings of our study are also consistent with widespread vascular damage since nearly all patients developed thromboses, 27 had hemorrhagic infarctions, and 71 had an increased D-dimer level. Furthermore, 12 persons developed PE. This finding elucidates a more specific aspect of COVID-19’s pathophysiology that damages the vasculature in multiple organ systems. Bryce et al. [19] found increased thromboembolic events in patients compared with the general population and multifocal acute infarctions and microthrombi in both venous and arterial vessels. This finding further proves the necessity for rapid anticoagulation in most COVID-19 patients with a severe clinical course.

The patient group in this study primarily consisted of the elderly with one or more severe co-morbidities. Our study is similar to other studies on autopsy research [17,19,29]. Autopsy studies are likely to skew towards multi-morbid and old populations, as these patients are more likely to die during the disease course [30,31]. This finding poses a challenge in evaluating certain aspects of COVID-19 pathophysiology. While microthrombi and thromboembolic events are common in the disease course, a large group of patients had factors that already facilitated these, e.g., diabetes mellitus [32], adiposity [33], or atherosclerosis in the coronary arteries. Caution is recommended in extrapolating clinical and morphological characteristics from autopsy patients and using them as references for the general population.

The inflammatory marker CRP levels were elevated in all but two cases. Furthermore, the mean CRP level was substantially increased, indicating a severe disease course in most patients, as this clinical parameter has been shown to correlate with disease severity in multiple studies [34,35]. However, histologically, a significant cellular inflammatory reaction was not seen since most of the patients only presented a sparse inflammatory infiltrate of lymphocytes and macrophages. This finding is consistent with research in 2020 by Pedersen and Ho [35], who concluded that cytokine and chemokine levels are markedly up-regulated and correlate with disease severity in COVID-19, whereas CD4+ and CD8+ T cell counts are significantly decreased in these same patients. The infection likely causes a cytokine storm responsible for T-cell depletion, leading to the sparse inflammatory infiltrate seen in most COVID-19 lung autopsy studies [17,18,19,29].

Lung coinfection is another cause of significant morbidity in viral pneumonia patients. In our study, only three cases were clinically confirmed, and 10 had histological signs of coinfection. Severe COVID-19 patients commonly require long stays in intensive care units and must be intubated, which are both significant risk factors for hospital-acquired pneumonia [36]. Other studies have shown superimposed acute pneumonia to be more common [19]. Therefore, it is important to recognize the possibility of coinfection in patients with an acute onset of new respiratory symptoms.

This study is limited by its focus on lung pathology alone. Bryce et al. [19] described numerous pathological changes in the cardiovascular, hematolymphoid, and central nervous systems, and less prominent changes in the genitourinary, gastrointestinal, hepatobiliary, and endocrine systems. Histological changes in the heart primarily consisted of myocyte hypertrophy and interstitial fibrosis consistent with pre-existing chronic conditions. This finding is not surprising, as their study’s cohort of patients had a median age of 68 and multiple co-morbidities. Nevertheless, COVID-19 likely causes significant changes in multiple organ systems. Clinical studies have shown abnormal heart function during infection and numerous months after initial diagnosis. Bergamaschi et al. [37] reported abnormal ECG findings at admission of 106 out of 216 patients and showed that ECG abnormalities correlated with intrahospital all-cause mortality and the need for invasive mechanical ventilation. Gherbesi et al. [38] evaluated 40 young adults with echocardiography at least 3 months after diagnosis and showed left ventricular deformation in 30% of them. This finding was despite patients having an asymptomatic or mildly symptomatic disease course. Further study is required on cardiovascular morphology in COVID-19 patients to understand its pathogenesis.

The COVID-19 pandemic is the most important event in recent memory. Its effects on medicine, science, and people’s daily lives cannot be understated. The pandemic will continue to be the most important factor determining social order and professional and hospital life in the foreseeable future; therefore, any attempt to fully understand the fundamentals of COVID-19 is valuable. Our goal for this study was to analyze morphological findings in lung tissues to increase the amount of research performed on this topic worldwide. There is a substantial research requirement to analyze histological data and their correlation with clinical findings in both Latvia and the world. Further research is required to fully understand COVID-19, preferably with communication and cooperation between pathologists and clinicians.

## Figures and Tables

**Figure 1 diagnostics-12-02808-f001:**
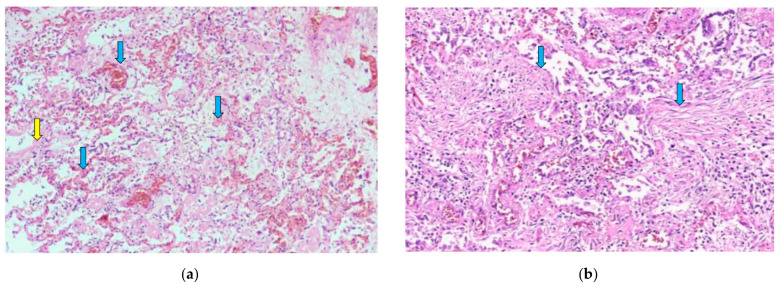
(**a**) Diffuse alveolar damage, exudative phase. Hyaline membranes (yellow arrow) and a sparse inflammatory infiltrate with mostly macrophages and lymphocytes are seen. Capillary thromboses (blue arrows) and some hemorrhaging are also visible. H & E stain; magnification ×40. (**b**) Diffuse alveolar damage, proliferative phase. Alveolar walls are thickened, and fibrous polyps can be seen in the lumen of alveoli (blue arrows). H & E stain; magnification ×100.

**Figure 2 diagnostics-12-02808-f002:**
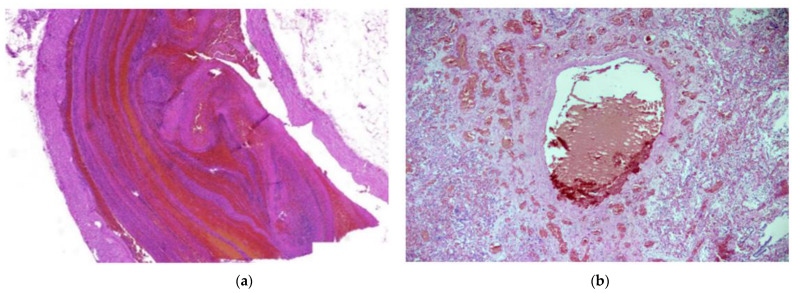
(**a**) Pulmonary thrombosis. An elongated thrombus filling almost the entire lumen of an artery can be seen. H & E stain; magnification ×30. (**b**) Thrombus in a large vessel with smaller vessel thromboses surrounding it. H & E stain; magnification ×40.

**Figure 3 diagnostics-12-02808-f003:**
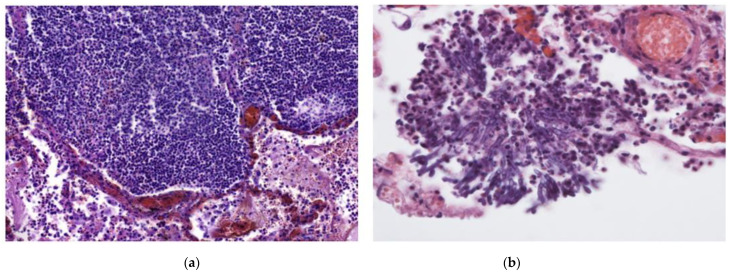
(**a**) In addition to DAD, this patient had an intense, focal inflammatory infiltrate primarily consisting of lymphocytes and foamy macrophages. H & E stain; magnification ×200. (**b**) Fungal hyphae found in the same patient as (**a**). Fungal co-infection explains the dense inflammatory infiltrate. H & E stain; magnification ×400.

**Table 1 diagnostics-12-02808-t001:** Summary of clinical parameters.

Parameter, Unit of Measurement	Value
Gender ratio (M:F)	56:32
Mean age ± SD, years (range)	62.2 ± 15.5 (22–94)
Mean length of hospital stay ± SD, days (range)	10.4 ± 9.8 (0–39)
Mean length of disease ± SD, days (range)	15.5 ± 10.1 (0–47)
Leukocytosis, count (frequency, %; 95% CI)	61/84 (72.6%; 95% CI 62.2–81.1)
Mean leukocyte count ± SD, 103/μL (range)	14.6 ± 8.9 (0.7–51.5)
Increased CRP level, count (frequency, %; 95% CI)	83/85 (97.7%; 95% CI 91.3–99.9)
Mean CRP level ± SD, mg/L (range)	177.2 ± 126.3 (0.8–505.4)
Increased D-dimer level, count (frequency, %; 95% CI)	71/76 (93.4%; 95% CI 85.2–98.0)
Mean D-dimer level ± SD, μg/mL (range)	9.1 ± 12.5 (0.4–71.3)
**Co-Morbidity**	**Count (Frequency, %; 95% CI)**
Coronary heart disease	49/88 (55.7%; 95% CI 36.5–56.9)
Primary arterial hypertension	41/88 (46.6%; 95% CI 36.5–56.9)
Adiposity	41/88 (46.6%; 95% CI 36.5–56.9
Diabetes mellitus	34/88 (38.6%; 95% CI 29.1–49.1)
Chronic kidney disease	24/88 (27.3%; 95% CI 19.0–37.4)
Chronic obstructive pulmonary disease	3/88 (3.4%; 95% CI 0.7–9.9)
Schizophrenia	2/88 (2.2%; 95% CI 0.1–8.4)
Bronchial asthma	1/88 (1.1%; 95% CI 0.0–6.8)
HIV infection	1/88 (1.1%; 95% CI 0.0–6.8)

Abbreviations: M, male; F, female; CRP, C-reactive protein; SD, standard deviation; CI, confidence interval.

**Table 2 diagnostics-12-02808-t002:** Summary of morphological findings in lung tissue samples.

Morphological Parameter	Count (Frequency, %; CI 95%)
Diffuse alveolar damage, exudative phase	38/88 (43.2%; 95% CI 33.3–53.6)
Diffuse alveolar damage, proliferative phase	45/88 (51.1%; 95% CI 40.8–61.3)
Vascular damage (hemorrhage, micro thromboembolism)	70/88 (79.5%; 95% CI 69.9–86.7)
Vessel (small, medium, large caliber) thromboses	83/88 (94.3%; 95% CI 87.0–97.9)
Fibrosis	52/88 (59.1%; 95% CI 48.6–68.8)
Hemorrhagic infarction	27/88 (30.7%; 95% CI 22.0–41.0)
Pulmonary embolism	12/88 (13.6%; 95% CI 7.8–22.5)
**Inflammatory Cells, Density of Infiltration**	
Neutrophil leukocytes	
Absent	47/88 (53.4%; 95% CI 43.1–63.4)
Scant	23/88 (26.1%; 95% CI 18.0–36.2)
Intermediate	9 (10.2%; 95% CI 5.2–18.5)
High	9 (10.2%; 95% CI 5.2–18.5)
Macrophages	
Absent	2/88 (2.2%; 95% CI 0.1–8.4)
Scant	33/88 (37.5%; 95% CI 28.1–48.0)
Intermediate	42/88 (47.7%; 95% CI 37.6–58.0)
High	11/88 (12.5%; 95% CI 7.0–21.2)
Lymphocytes	
Absent	5/88 (5.7%; 95% CI 2.1–12.9)
Scant	47/88 (53.4%; 95% CI 43.1–63.5)
Intermediate	26/88 (29.5%; 95% CI 21.0–39.8)
High	10/88 (11.4%; 95% CI 6.1–19.8)
Multinucleated giant cells	60/88 (68.2%; 95% CI 57.8–77.0)

Abbreviations: CI, confidence interval.

## Data Availability

The data are available only by request.

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
