# Peer review of "Pathological Changes in the Lungs of Patients with a Lethal COVID-19 Clinical Course"

_diagnostics, 2022, doi:10.3390/diagnostics12112808_

Round 1

Reviewer 1 Report

The paper reports the most significant histopathological lung  findings in a case series of COVID autopsies in Latvia. It may be quite interesting as morphological patterns characteristic are still partly to be explored. However, some changes are needed. First of all, the Introduction is confusing and it could be better rewritten. Data regarding COVID incidence and autopsy rate in Latvia are superfluous. I suggest deleting them and adding general data on COVID autospies worldwide (see the numerous literature review on the topic, among others, Maiese et al, etc).  

Lines 118-120 The meaning is not clear; please clarify what Authors mean. 

Fig. 1.b. I'not sure that the arrows indicates a DAD proliferative phase. I suggest replacing the photo with a better one. 

Author Response

Thank you for your comments and suggestions. The Authors have made changes to the manuscript as requested. Please see the attachment.

Reviewer 2 Report

The authors examined the morphology of the lungs in 88 patients that died due to COVID-19 in Latvia thus increasing the amount of data available about the histological characteristics of SARS-CoV-2-induced disease. They concludede that the histological findings of marked diffuse alveolar damage, widespread vascular lesions and active thrombosis can be considered representative of severe COVID-19.

This is a very interesting manuscript.

I suggest to include in this article the following papers:

1) DOI: 10.1111/anec.12815 2) DOI: 10.1111/echo.15431

Author Response

Thank you for your comments and suggestions. The authors have made changes to the manuscript as requested. The manuscript has also been edited by MDPI English Editing. 

Point 1: The authors examined the morphology of the lungs in 88 patients that died due to COVID-19 in Latvia thus increasing the amount of data available about the histological characteristics of SARS-CoV-2-induced disease. They concludede that the histological findings of marked diffuse alveolar damage, widespread vascular lesions and active thrombosis can be considered representative of severe COVID-19.

This is a very interesting manuscript.

I suggest to include in this article the following papers:

1) DOI: 10.1111/anec.12815 2) DOI: 10.1111/echo.15431

Response 1: The authors have included a paragraph referencing the above mentioned articles in the Discussion section of the manuscript:

“This study is limited by its focus on lung pathology alone. Bryce et al. described numerous pathological changes in the cardiovascular, hematolymphoid, and central nervous systems, and less prominent changes in the genitourinary, gastrointestinal, hepatobiliary, and endocrine systems. Histological changes in the heart primarily consisted of myocyte hypertrophy and interstitial fibrosis consistent with pre-existing chronic conditions. This finding is not surprising as their study’s cohort of patients had a median age of 68 and multiple co-morbidities [19]. Nevertheless, COVID-19 likely causes significant changes in multiple organ systems. Clinical studies have shown abnormal heart function during infection and numerous months after initial diagnosis. Bergamaschi et al. reported abnormal ECG findings at admission of 106 out of 216 patients and showed that ECG abnormalities correlated with intrahospital all-cause mortality and the need for invasive mechanical ventilation [37]. Gherbesi et al. evaluated 40 young adults with echocardiography at least 3 months after diagnosis and showed left ventricular de-formation in 30% of them. This finding was despite patients having an asymptomatic or mildly symptomatic disease course [38]. Further study is required on cardiovascular morphology in COVID-19 patients to understand its pathogenesis.”

Point 1: Moderate English changes required.

Response 1: The authors have submitted the manuscript to MDPI English Editing and received the revised version and the English editing certificate.